# Enterotoxigenic *Escherichia coli* Heat-Stable Toxin and Ebola Virus Delta Peptide: Similarities and Differences

**DOI:** 10.3390/pathogens11020170

**Published:** 2022-01-27

**Authors:** Lilia I. Melnik, Robert F. Garry

**Affiliations:** 1Department of Microbiology and Immunology, Tulane University School of Medicine, New Orleans, LA 70112, USA; rfgarry@tulane.edu; 2Viral Hemorrhagic Fever Consortium, New Orleans, LA 70112, USA

**Keywords:** Ebola virus, delta peptide, enterotoxigenic *Escherichia coli*, STb, enterotoxin, diarrhea, mechanisms of fluid secretion

## Abstract

Enterotoxigenic *Escherichia coli* (ETEC) STb toxin exhibits striking structural similarity to Ebola virus (EBOV) delta peptide. Both ETEC and EBOV delta peptide are enterotoxins. Comparison of the structural and functional similarities and differences of these two toxins illuminates features that are important in induction of pathogenesis by a bacterial and viral pathogen.

## 1. Introduction

Viral and bacterial infections are the most frequent causes of diarrhea and can result in large outbreaks. Enterotoxigenic *Escherichia coli* (ETEC) causes yearly outbreaks of diarrhea in young children and farm animals. The hallmarks of the 2013–2016 West African Ebola outbreak were gastrointestinal symptoms such as vomiting and severe diarrhea. Diarrhea was a major predictor of fatal outcome in patients with confirmed Ebola virus disease (EVD). ETEC STb toxin exhibits striking structural similarity to Ebola virus (EBOV) delta peptide. STb is a 48-amino-acid peptide with two disulfide bonds that are required for enterotoxigenic activity. EBOV delta peptide, a 40-residue nonstructural peptide, is a viroporin and acts as an enterotoxin. The viroporin activity of EBOV delta peptide is dependent on the presence of a disulfide bond, but it is not required for the induction of diarrhea. In this review, we analyze structural and functional similarities and differences of these two toxins and highlight features that are important in induction of pathogenesis.

## 2. Heat-Stable Toxin II (STb)

### 2.1. Epidemiology

Enterotoxigenic *Escherichia coli* (ETEC) is a major cause of acute diarrhea in humans and farm animals. Populations that suffer from acute diarrheal disease due to ETEC infections are infants and children less than 5 years of age in developing countries and travelers within ETEC endemic areas [1,2,3,4]. ETEC produces heat-labile toxin (LT) and two types of heat-stable toxins (STs), STI (also known as STa) and STII (also known as STb) [5,6]. ETEC strains that produce LT and STa were identified as etiologic agents of diarrhea in animals and humans [5,7]. While STa causes diarrhea in humans, newborn piglets, and calves, STb, on the other hand, is known to cause diarrhea primarily in weaned pigs [8,9]. Reports describing STb as a causative agent of secretory diarrhea in humans showed that the nucleotide sequence of the STb gene from human isolates is similar to the STb gene found in porcine strains [10,11]. In a recent large multicenter study conducted in sub-Saharan Africa and South Asia, ETEC-producing STb was identified as one of four pathogens found in children of up to five years of age with less-severe diarrhea and moderate-to-severe diarrhea and was associated with increased risk of death [12,13]. Similar results were obtained in another study that performed community surveillance for diarrhea at multiple sites in South America, Africa, and Asia; STb was identified as one of the six most common pathogens in children at the age of two and was associated with episodes of dysentery, dehydration, or hospitalization [14].

ETEC infections are very common in farm animals, leading to substantial economic losses for this industry [15,16]. ETEC-producing STb causes diarrhea in neonatal and weaned pigs that can lead to growth retardation or result in mortality. A progressive increase in the number of STb-positive ETEC strains is observed in piglets older than one week [17]. ETEC-producing STb causes diarrhea in about 70 to 90% of weaned pigs [18]. Chapman and coworkers employed uni- and multiplex PCR methods to identify gene profiles in healthy and diarrheic pigs by targeting fifty-eight virulence genes associated with *E. coli* known to cause disease in humans and animals. The STb gene was associated with diarrhea in 25% of neonatal pigs and 81.3% of weaned pigs [19]. Multiple ETEC genotypes have also been identified in pigs without any clinical signs of diarrhea. A study on subclinical ETEC infections conducted in Argentina by Moredo and coworkers revealed a high prevalence (97.5%) of the *estB* gene in ETEC strains found in neonatal and post-weaning pigs that were not exhibiting symptoms of diarrhea [20]. Diarrhea resulting from ETEC strains producing STb was also observed in calves and chickens, but with lower frequency compared to pigs [17].

### 2.2. Structure

STb enterotoxin produced by ETEC is encoded by a plasmid-associated *estB* gene that is highly conserved in ETEC isolates around the world [21,22,23]. However, a variant of STb with a point mutation at position twelve (H12N) was detected in Canada and Hungary [24]. STb is synthesized as a 71-amino-acid precursor protein followed by a single processing event that produces an N-terminal 23-amino-acid signal peptide and a 48-amino-acid mature peptide. The signal peptide is removed during its translocation through the *E. coli* inner membrane. The mature peptide containing four cysteines is translocated to the periplasmic space [22,25] where it undergoes disulfide bond formation and is secreted into extracellular space without further modifications [25,26,27]. The disulfide bonds between C10 and C48 and between C21 and C36 stabilize the mature peptide in a configuration with two antiparallel α-helices (C10-K22 and G38-A44) separated by a loop [26,28,29] (Figure 1A,B). Disulfide bonds stabilize STb from proteolysis in a periplasmic space [25,26]. Reduction of cysteine bonds results in a random coil-like structure [29,30]. Importantly, both disulfide bonds are indispensable for biological activity of STb [30,31]. Mutants generated through site-directed mutagenesis of the cysteine residues showed that the loss of either disulfide bond results in loss of biological activity when tested in rat intestinal loops [30]. Substitutions of positively charged lysine residues at positions 18 (K18G), 22 (K22L), 23 (K23T), and 46 (K46S) resulted in reduced secretory activity of STb in mouse intestinal loop assay. The highest inhibition of STb-induced fluid secretion was achieved through the substitution of K22 and K23, indicating the importance of these amino acids in biological activity of STb [32]. Additionally, charged residues of STb, R29, and D30 were shown to be involved in enterotoxigenic activity when tested in rat intestinal loop assay [26]. Site-directed mutagenesis identified residues important for enterotoxicity and attachment of STb to its receptor. When residues located in the hydrophobic helix were substituted with basic and polar residues (F37K, I41S, and M42S), enterotoxicity and receptor binding decreased more than sixfold. Mutants of basic residues (K22A, K23A, and R29A) showed reduced receptor binding and reduced enterotoxicity in rat ligated loop assay. While substitutions of negatively charged residue D30 (D30A and D30V) reduced enterotoxicity by half, binding of these STb mutants to its receptor increased enterotoxicity by twofold [33]. Labrie and coworkers studied protein–protein interaction of STb toxin and reported that STb forms hexamers and heptamers in a temperature- and receptor-independent manner [34]. The functional receptor for STb is sulfatide [Gal(3-SO_4_)β1Cer], a glycosphingolipid that is present on the cell surface of the jejunal epithelium [35,36]. Importantly, residues F37, I41, and M42 are critical for the formation of hydrophobic bonds between C-terminal α-helices to stabilize the tertiary structure of STb (Figure 1B and Table 1) [34].

### 2.3. Pore Formation

Most bacterial toxins oligomerize to permeate cellular membranes [39] and are termed pore-forming toxins (PFTs). Pore-forming toxins are classified into two classes based on the secondary structure that forms the pore. The majority of bacterial toxins are β-PFTs that form β-barrels assembled from individual β-sheets. PFTs that form pores consisting of amphipathic α-helices are called α-PFTs [40]. STb monomer contains two α-helices that form hexamers and heptamers. STb formed nonspecific pores in membrane vesicles isolated from porcine jejunum. The disulfide bridges and residues K22, K23, and M42 that are essential for STb’s enterotoxicity and oligomerization of STb are also indispensable for the ability of STb to permeabilize cellular membranes (Figure 1B and Table 1) [41]. STb permeabilized membranes of NIH-3T3 cells and was internalized by the cells. Initially, STb was found scattered throughout the cytoplasm and later was detected in mitochondria that lead to mitochondria hyperpolarization [42], which might be an indication of early stages of apoptosis as shown by previous studies [43,44]. Treatment with STb resulted in histological changes such as cytoplasm granularization and morphological alterations associated with apoptosis including enlargement of the nucleus and membrane budding [42].

### 2.4. Enterotoxigenic Activity of STb

By the late 1970s, it was established that ETEC produces two enterotoxins: LT and ST. Burgess and coworkers [8] reported that during purification of ST secreted from P16, a porcine strain of *E. coli*, two heat-stable toxins were produced. The first toxin, STa, is methanol-soluble and partially heat-stable; the second toxin, STb, is methanol-insoluble and heat-stable [8]. The two toxins were differentially active in animal models. STa was active in neonatal pig ligated loop assay and suckling mouse assay but inactive in weaned pig ligated loop assay. STb induced fluid secretion in ligated intestinal loops of weaned pigs and rabbits but did not act as an enterotoxin in the suckling mouse assay [8,45]. STb-positive strains (T2 and UK/A) tested in ileal loops of piglets induced a diarrheal response [46]. Subsequent work by Kennedy and coworkers [9] confirmed that STb did not produce secretory responses in suckling mice and ligated loops of rats and rabbits but induced fluid secretion in intestinal loops of weaned pigs. Unlike previous studies where the secretory responses were assessed at one time point post-injection (16–18 h) [8,45], Kennedy and coworkers evaluated the ability of STb to induce fluid secretion at several time points post-injection into ligated loops (0.5, 3, 6, and 16 h). Maximal STb-induced fluid secretion was observed at 3 to 6 h post-injection, and no obvious changes to villous and crypt cell morphology were detected [9]. Another study that evaluated the time course of STb-induced diarrheal response was performed by Rose and coworkers [47], wherein the STb-induced diarrheal response was evaluated 2 h post-injection. A small number of pigs were euthanized at 15, 30, 60, and 120 min. The highest secretory response was observed 2 h after injection of STb into porcine ligated loops. Out of twenty 2-to-3-week-old pigs, three pigs did not produce a diarrheal response after STb injection into the loops [47]. The discrepancy in diarrheal responses in pigs was also observed and addressed by Whipp and coworkers [48] and was attributed to susceptibility of STb to degradation by trypsin present in jejunal lumen of weaned pigs. It has been shown previously that trypsin acts primarily on lysine and arginine residues [49]. When jejunal lumen was prerinsed with saline and then subjected to STb toxin, fluid secretion was observed in all animals tested [48].

Initial studies evaluating the enterotoxigenic activity of STb employed suckling mouse assays and ligated loop assays in neonatal pigs, weaned pigs, rats, rabbits, and lambs. Consistent responses were observed in weaned pigs only, and it was believed that STb-induced diarrheal responses are pig-specific [8,9,45,47]. Fluid secretion in ligated loops of rabbits was also reported by Burgess and coworkers, but it was considerably weaker compared to diarrheal responses seen in porcine ligated loops [8]. Attempts to demonstrate the ability of STb to elicit secretory responses in mice and rats were not successful [9,45], but when endogenous protease activity was blocked by protease inhibitor, STb induced fluid secretion in suckling mice and ligated loops of mice, rats, calves, and rabbits [50,51,52].

### 2.5. Histological Changes

The first report that STb induces histological changes such as partial villous atrophy and crypt hyperplasia in pig jejunum [53] was followed by findings that STb causes alterations in porcine jejunal epithelium, including shortening of villi, increase in number of sloughed cells, loss of villous absorptive cells, and partial villous atrophy, that lead to changes in absorption [48,54]. Rose and coworkers examined the STb-induced histological changes in pig, rabbit, and lamb ligated ileal loops. Histological alterations were preceded by STb-induced secretory responses and included changes in epithelial morphology from columnar to cuboidal or squamous. In addition, the presence of gaps in the epithelium was observed. These changes in villous morphology and epithelial layer appearance were observed 2 h after pig and lamb intestines were exposed to STb. These abnormalities were not observed in rabbits [47]. When purified STb was injected into washed murine intestinal loops, fluid secretion was observed 3 h after surgery. Histological examination did not show any cellular damage or inflammation, but dilation of capillaries and decrease in thickness of lamina propria were observed [52].

### 2.6. Mechanism of STb

ETEC pathogenesis begins with adhesion of the bacterium to the intestinal epithelium mediated by colonization factors and release of enterotoxins that subsequently bind their receptors [55]. STb binds to its receptor, sulfatide, through terminal oligosaccharide sequence [Gal(3-SO_4_)β1Cer] on sulfatide with the optimum binding pH of 5.8 [35,36,56]. Recently identified STb variant H12N was reported to bind the same receptor [57]. Once bound, STb is internalized [58], followed by a dose-dependent influx of extracellular Ca^2+^ through the activation of pertussis toxin-sensitive GTP-binding regulatory protein and receptor-operated Ca^2+^ channel [59,60]. Intracellular increase in Ca^2+^ levels in response to STb leads to binding of Ca^2+^ to its receptor calmodulin, which results in the formation of Ca^2+^-calmodulin complex that promotes activation of calmodulin-dependent protein kinase II (CAMKII) [61]. STb caused the release of prostaglandin E2 (PGE2) into fluid accumulated in STb-treated mouse intestinal loops. Treatment with aspirin and indomethacin reduced the effect of STb, suggesting the involvement of PGE2 in the mechanism of STb-induced fluid secretion. STb did not alter levels of intestinal cGMP and cAMP after the 3-h treatment of mouse intestinal loops. These results indicate that STb induces diarrhea by a guanylate- and adenylate-cyclase-independent mechanism [52,62]. STb treatment of rat ligated intestinal loops resulted in dose-dependent increase in fluid secretion, accompanied by an increase in levels of PGE2 and serotonin (5-HT). Pretreatment with ketanserin, a 5-HT_2_ receptor inhibitor, reduced STb-induced secretion and the level of PGE2 in rat intestinal secretions, suggesting that a subset of PGE2 is synthesized as the result of serotonin receptor stimulation. Similarly, indomethacin, a COX-2 inhibitor, significantly reduced STb-induced diarrheal response and had no effect on the levels of 5-HT. Treatment of rat ligated intestinal loops with both inhibitors had a synergistic effect on STb-induced fluid secretion [63]. STb treatment of pig intestinal loops stimulated the release of PGE2 and 5-HT by the intestinal mucosa, suggesting the involvement of enteric nervous system in STb-induced diarrhea [62]. While STb stimulates arachidonic acid release and metabolism, resulting in PGE2 synthesis, 5-HT possibly promotes PGE2 synthesis through phospholipase A_2_ (Figure 2) [63,64,65].

Tight junctions are proteins that hold adjacent enterocytes together, forming intestinal mucosa, and control the transport of solutes, ions, and water. Treatment with STb showed an increase in paracellular permeability in T84 human colon cells compared to untreated cells or cells treated with the STb mutant D30V. STb treatment was associated with F-actin filament dissolution and condensation accompanied by redistribution and fragmentation of ZO-1, claudin-1, and occludin. These changes were also observed after treatment of cells with the synthetic peptide GFLGVRDG, which corresponds to STb residues 24 to 31 [66]. Additionally, STb treatment of T84 cells resulted in dephosphorylation of claudin-1 and its displacement from tight junctions to the cytoplasm [67].

## 3. Ebola Virus Delta Peptide

### 3.1. Epidemiology of Ebola Virus

Ebola virus (EBOV) (family *Filoviridae*) infects humans and non-human primates. Outbreaks of Ebola virus disease (EVD) in humans are associated with high case fatality rates (CFR). The symptoms of EVD vary between individuals and can include hemorrhagic symptoms such as maculopapular rash, mucosal hemorrhage, and petechiae. The first known outbreak of EVD occurred in 1976 in the Democratic Republic of the Congo (DRC) with the CFR of 88% [68], followed by three other outbreaks a few years apart from each other [69,70]. The common feature of these outbreaks was that the infections were documented in humans only. No infections of wild animals were observed [71].

Ebola virus reappeared in the period between 1994 and 1997 primarily in Gabon, resulting in four epidemics of EVD, and was associated with deaths among gorillas, chimpanzees, and bush pigs in the surrounding forested areas. Contact with animals was believed to be the source of the infection that led to these outbreaks in a human population [69]. Subsequent outbreaks of EVD in the early 2000s resulted from contact of humans with gorillas, chimpanzees, monkeys, and carcasses of wild animals [72,73,74]. Between 1994 and 2003, forty-four separate outbreaks of morbidity and mortality in wild animals were documented around Gabon and the northwestern DRC, with sixteen outbreaks out of those forty-four coinciding with human EVD outbreaks both temporally and spatially. During observational studies conducted in this area, carcasses of different animals were found. Laboratory confirmation of EBOV infection for the carcasses of gorillas, chimpanzees, and duiker antelope and epidemiological investigations showed that Ebola outbreaks in the human population were the result of multiple introductions of the virus from different EBOV-infected animals [73,74]. Witnesses also described seeing dying mandrills, chimpanzees, black colobuses, and bush pigs exhibiting symptoms such as vomiting and/or diarrhea, severe diarrhea, and emaciation [71]. Similar symptoms were observed in non-human primates experimentally infected with EBOV in the laboratory [75,76].

The largest EVD outbreak occurred in 2013–2016 in Guinea, Liberia, and Sierra Leone. One of the hallmarks of the 2013–2016 West African outbreak was that hemorrhagic symptoms characteristic of EVD reported during earlier outbreaks were observed less frequently. Instead, gastrointestinal symptoms such as abdominal pain, nausea, vomiting, and diarrhea were more prominent in patients with EVD [77,78,79,80,81,82]. Diarrhea was associated with mortality in patients with EVD [77,83,84,85]. Up to 83% of patients who succumbed to Ebola suffered from diarrhea [84,86,87,88,89].

Ebola virus causes various pathologies in humans and animals that include gastrointestinal disease. The role of the enterotoxigenic EBOV delta peptide [90] has yet to be completely explored and may inform future investigations.

### 3.2. Structure

EBOV delta peptide is encoded by the GP gene of EBOV as the result of transcriptional editing. Transcriptional editing of the GP gene can result from the addition of one or more untemplated adenosines to the mRNA editing site, resulting in three different mRNAs that code for three protein products: full-length glycoprotein (GP), soluble glycoprotein (sGP), and second soluble glycoprotein (ssGP) [91,92,93,94,95,96,97]. The primary product of the GP gene is sGP [98], which is translated from the unedited GP mRNA species to produce a pre-soluble glycoprotein (pre-sGP). Pre-sGP undergoes co- and post-translational processing, including cleavage of signal peptide, glycosylation, and proteolytic cleavage by furin at the alternate site from pre-GP to produce the N-glycosylated sGP disulfide-linked homodimer and O-glycosylated and sialylated C-terminal sGP fragment termed the delta peptide [94,99,100]. While sGP is released from the cells after proteolysis, delta peptide remains cell-associated and is released extracellularly in considerably smaller amounts compared to sGP [94].

EBOV delta peptide is a 40-amino-acid-long nonstructural peptide. It consists of an N-terminal domain variable across different filoviruses and a more conserved C-terminal domain. Garry and Gallaher noticed that the C-terminal portion of EBOV delta peptide (amino acids 18–38) is highly conserved throughout Ebola virus species, where 9/21 amino acids are identical. A high level of conservation of cationic amino acids such as lysine, arginine, and histidine has been observed between delta peptides from Ebola virus species and a more distantly related cuevavirus. Presence of hydrophobic (7/23) together with charged and polar residues (8/23) in the C-terminal portion of delta peptide indicated that delta peptide could form an amphipathic α-helix, which was supported by various computer modeling methods (Figure 3A,B). Additionally, the presence of two conserved cysteine residues in delta peptide sequences of Ebola viruses and cuevavirus strongly suggested the possibility that delta peptide can form a biologically active secondary structure containing the disulfide bond. EBOV delta peptide was also tested for its propensity to interact with lipid membranes by using the Wimley–White interfacial hydrophobicity scale (WWIHS), an experimentally determined hydrophobicity scale that predicts regions of proteins that may interact with membranes [101]. EBOV delta peptide had a positive score of 3.66, which indicates that it has a high propensity to interface with lipid membranes [102]. Garry and Gallaher hypothesized that delta peptide could oligomerize as a tetramer and form a pore with hydrophilic positively charged amino acids facing inward and hydrophobic amino acids interacting with the lipid environment of the target membrane. A pore lined with positively charged residues would allow chloride ion flux through the cell membrane of enterocytes that could result in diarrhea [102]. All-atom molecular dynamics (MD) simulations performed by Pokhrel and colleagues showed that Ebola virus delta peptide could form pentameric pores selective to chloride ions [103]. Structural studies such as NMR are needed to elucidate the structure of EBOV delta peptide.

### 3.3. Viroporin Activity

Modeling studies performed by Garry and Gallaher suggested that EBOV delta peptide can permeabilize lipid membranes. We investigated the viroporin activity of EBOV delta peptide in vitro. Synthetic peptides corresponding to the full-length of EBOV peptide (E40_ox_) and the conserved C-terminal portion (E23_ox_) in oxidized form displayed cytotoxic activity and the ability to permeabilize cellular membranes. Both peptides increased ion permeability across cellular membranes and synthetic bilayers, confirming that EBOV delta peptide acts as a viroporin. The ability of EBOV delta peptide to permeabilize membranes is entirely dependent on a disulfide bond between the two cysteines [104]. When the cysteine residues were substituted with alanine residues (C→A), the cytotoxic effect of E23_ox_ on Vero cells was reduced twofold. Substitution of lysine residues with glutamine (K→Q) reduced the cytotoxic effect of E23_ox_ twofold and cytotoxic effect of E40_ox_ fourfold. Replacement of lysine–glutamic acid sequence with glutamic acid–lysine pair (KE→EK) in E23_ox_ and E40_ox_ reduced cytotoxicity twofold. Replacement of lysine–glutamic acid sequence with two glutamine residues (KQ→QQ) in E40_ox_ reduced delta peptide’s cytotoxicity twofold. Fourfold reduction in E40_ox_ -induced cytotoxicity was observed as the result of substitution of arginine residues with glutamic acid (R→E) (unpublished data). These data suggest the importance of cysteines and basic amino acids in cytotoxic activity of EBOV delta peptide.

Viroporins play an important role in viral pathogenesis and are major therapeutic targets. They are usually small hydrophobic proteins containing a few amphipathic helices that upon oligomerization form pores in cellular membranes. Additionally, viroporins change permeability of cellular membranes to ions and to small molecules [105]. For instance, matrix protein M2 of influenza virus A, a proton channel, aids the virus during the uncoating step by allowing entry of protons from the late endosome into the influenza A virion, which leads to acidification of the interior of the virion, dissociation of M1 matrix protein from RNP, and subsequent release of RNP from the virion [106,107,108,109]. HIV-1 viral protein U (Vpu) downregulates the expression of HIV-1 receptor CD4 in the ER, which allows processing and transport of mature viral proteins to the site of assembly of newly made viral particles. Furthermore, Vpu antagonizes tetherin, which cross-links newly made viral particles on the surfaces of the infected cells, resulting in their efficient release [110,111,112]. HIV-1 lentivirus lytic peptide 1 (LLP-1) is a pore-former and induces neurotoxicity associated with cognitive deficits in AIDS patients [113,114].

Another viroporin, and, until recently, the only known viral enterotoxin, is non-structural protein 4 (NSP4) of rotavirus, which causes large yearly outbreaks of diarrhea in young children worldwide [115]. NSP4 forms tetramers and pentamers consisting of amphipathic α-helices rich in lysine residues that localize and form pores in membranes of endoplasmic reticulum (ER) of infected cells and change intracellular Ca^2+^ levels [116,117,118]. Ball and coworkers demonstrated that NSP4 and a synthetic peptide corresponding to residues 114 to 135 of the protein induce diarrhea in mice in an age-dependent and dose-dependent manner [119]. NSP4 increases intestinal permeability and induces chloride secretion through the intracellular increase in Ca^2+^ levels due to Ca^2+^ efflux from the endoplasmic reticulum (ER) and Ca^2+^ influx through the plasma membrane [120,121,122,123,124,125]. NSP4 has two domains that disrupt Ca^2+^ homeostasis and play an important role in induction of vomiting and diarrhea: NSP4 amino acid domain 47–90 and NSP4 amino acid domain 144–135. NSP4 amino acid domain 47–90, the viroporin domain of NSP4, forms a Ca^2+^-permeable ion channel in the ER that allows the persistent efflux of Ca^2+^ from the ER into the cytosol [125,126,127,128]. Increase in cytosolic calcium in rotavirus-infected cells activates pro-survival pathways that include phosphoinositide 3-kinase (PI3K) as well as pathways that promote replication of rotavirus and assembly of newly made viral particles [129]. Elevated Ca^2+^ levels in the cytosol lead to mitochondrial Ca^2+^ uptake, resulting in increase in ATP synthesis. However, continuous elevation of cytosolic Ca^2+^ levels ultimately results in activation of pro-apoptotic pathways [130], which could be delayed through the activation of PI3K in the early stages of infection [131]. During late stages of rotavirus infection, a decrease in ATP production and Ca^2+^-induced signaling leads to cell damage and cell death [131,132]. NSP4 amino acid domain 144–135, the enterotoxin domain of NSP4, is responsible for chloride secretion through transient increase in intracellular calcium that requires receptor-mediated-phospholipase-C-dependent production of 1,4,5-triphosphate (IP_3_) [120,133,134]. Gallaher and Garry in their modeling study [102] discovered a lysine-rich motif at the C-terminus of Ebola virus delta peptide that displays a high predicted structural similarity to the cytolytic peptide of NSP4.

### 3.4. Enterotoxigenic Activity

The ability of EBOV delta peptide to induce a diarrheal response was tested in a previously established murine ligated ileal loop model of diarrheal disease [135]. This model was originally used to study *V. cholerae* and was adopted for the study of the delta peptide. Melnik and coworkers investigated the role of delta peptide in EBOV delta peptide-induced diarrhea by employing this model in six-weeks-old BALB/c mice (13–22 g), which are widely used to study gastrointestinal diseases. In these experiments, a 2 cm loop of distal ileum was ligated and injected with 100 μL of either vehicle control or different concentrations of synthetic peptides corresponding to the C-terminal portion of EBOV delta peptide in oxidized form (E23_ox_). *V. cholerae* was used as a positive control. The data were collected at 6, 9, 12, and 24 h after surgery to establish the time course of delta peptide activity. Diarrheal response was determined by the ratio of loop weight over length (g/cm). Moderate fluid secretion accompanied by some increase in the width of the loop was observed at 6 h after injection of 10–100 μM of E23_ox_. The peak of E23_ox_-induced fluid secretion with the dramatic distention of the loops was observed 9 to 12 h post-injection. Additionally, during these time points, fluid-filled loops appeared highly vascularized with cellular debris collected inside the loops, which can be attributed to delta peptide-induced cytotoxicity observed in our earlier in vitro studies [104]. The diarrheal response at these time points was concentration-dependent, with the highest loop ratios resulting from the injection of 50 and 100 μM delta peptide. The peak of diarrheal response induced by E23_ox_ was observed earlier and was more extensive than fluid secretion induced by *V. cholerae*. Loops collected 24 h post-injection with E23_ox_ resembled the loops injected with vehicle control.

Full-length EBOV delta peptide (E40_ox_) and delta peptide serum cleavage products E17_ox_ and E15_ox_, which have two cysteine residues and can form a disulfide bond [104], were also tested for their ability to induce fluid secretion 12 h after treatment. Both E40_ox_ and E17_ox_ induced fluid secretion that was comparable to previously tested E23_ox_. The most stable serum cleavage product, E15_ox_, did not induce fluid secretion. However, E23_alk_, the C-terminal portion of EBOV delta peptide that does not contain the disulfide bond due to sulfhydryl alkylation, induced a moderate diarrheal response compared to E23_ox_. These results suggest that the secondary structure of Ebola virus delta peptide is important for induction of fluid secretion [90].

### 3.5. Histological Changes

EBOV delta peptide-induced fluid secretion was accompanied by striking structural changes to the intestinal mucosa. Tissues from loops injected with vehicle control, *V. cholerae*, or Ebola virus delta peptide were stained with period acid–Schiff (PAS) reagent and counterstained with hematoxylin to examine histological changes due to different treatments. PAS reagent detects mucins in goblet cells. Mucin secretion from goblet cells creates a mucus layer responsible for protection of the intestinal epithelium from various pathogens and toxins. Tissues from loops injected with vehicle control showed long and narrow villi with a high number of PAS-positive goblet cells compactly lining the interior of the ileal tissue. The treatment with E23_ox_ for 9 to 12 h resulted in the reduction of the number of PAS-positive goblet cells and dramatic changes in the architecture of the small intestine that were dose- and time-dependent, with the peak coinciding with the peak of E23_ox_-induced diarrhea. Architectural changes resulting from E23_ox_ treatment included reduction in the length of the villi, destruction of villi, and increase in crypt length. These architectural changes were mostly due to the dramatic stretching of the ileum as the result of E23_ox_-induced fluid secretion. As the diarrheal response due to delta peptide treatment was subsiding at 24 h after treatment, these architectural changes were reversed as the tissue was returning to its original shape. More importantly, 24 h after treatment with E23_ox_, intestinal mucosa was undergoing a repair process characterized by the appearance of villi with blunt tips reminiscent of newly grown villi, decrease in crypt length, and the appearance of mucin-containing goblet cells [90].

### 3.6. Mechanism of Ebola Virus Delta-Peptide-Induced Diarrhea

A healthy gastrointestinal tract is supported by numerous transporters, exchangers, and channels that function together to maintain fluid balance between absorption and secretion. This balance is disrupted by pathogens during enteric infections, resulting in either malabsorption, hypersecretion, or both, which are characteristic of diarrheal diseases. Oral rehydration therapy (ORT) is an important part of supportive care during pathogen-induced diarrheal disease. Successful ORT depends on the presence of functional and intact transporters and exchangers in enterocytes of intestinal mucosa. The villous epithelium of the small intestine utilizes Na^+^- glucose cotransporter SGLT-1, Na^+^/H^+^ exchanger NHE3, Na^+^- short chain fatty acid cotransporter SMCT1, and Cl^−^/HCO_3_^−^ exchanger DRA (down-regulated in adenoma) for absorption of ions and nutrients such as sugars, peptides, and fatty acids, followed by passive entry of water through the paracellular and transcellular routes [136]. Sodium absorption is the result of complex interchange of transportive processes powered by various exchangers and cotransporters located in the apical and basolateral membranes. Na^+^/K^+^ ATPase located in the basolateral membrane provides an electrochemical gradient for Na^+^ ion entry through the Na^+^- glucose cotransporter SGLT-1, Na^+^/H^+^ exchanger NHE3, and Na^+^- short chain fatty acid cotransporter SMCT1 located in the apical membrane. Na^+^/K^+^ ATPase is also responsible for the transport of Na^+^ ions from epithelial cells to the bloodstream [137,138]. RNA-seq analyses performed by Melnik and coworkers showed that E23_ox_ reduced the expression of *Slc5a*, which codes for SGLT-1, suggesting that the coupled transport of sodium and glucose might be impaired during Ebola infection [90]. It has been reported previously that NSP4 of rotavirus inhibits SGLT-1 in mouse colon [139] and in rabbit intestinal brush border membrane [140]. ORT during rotavirus-induced diarrhea is successful, and SGLT-1 continues to be functional [141]. Unlike EBOV delta peptide, rotavirus NSP4 causes only minimal damage to the intestinal mucosa that includes blunting of villi and a small number of lesions, leaving a sufficient number of SGLT-1 cotransporters undamaged [142,143]. Sodium/hydrogen exchanger NHE3 works in concert with chloride/bicarbonate exchanger DRA to allow absorption of electroneutral sodium chloride [144]. The expression of *Slc9a3r1*, which codes for NHE3, and *Slc26a3* and *Slc26a6*, which code for DRA exchangers, was downregulated as the result of treatment with E23_ox_, which might result in reduced absorption of electroneutral NaCl [90]. *Slc26a3* transcript encodes an exchanger that has the stoichiometry of two chloride ions and one bicarbonate ion, while *Slc26a6* encodes an exchanger that has been shown to allow the exchange of one chloride ion and two bicarbonates ions [145,146]. Mice that carry loss-of-function in *Slc26a3* gene have more severe chloride absorption deficiency than *Slc26a6*^−/−^ mice [147]. Loss-of-function mutation in *Slc26a3* gene in humans results in severe congenital chloride-losing diarrhea (CLD) [148]. Since E23_ox_ decreased the expression of both isoforms of *Slc26* gene that code for DRA, chloride absorption might be greatly reduced during EBOV-induced gastrointestinal disease due to the loss-of-function of *Slc26a6* isoform that cannot be compensated by *Slc26a3*. *Slc5a8* that codes for the Na^+^- dependent carboxylate transporter for small chain fatty acids, lactate, and pyruvate (SMCT1) is expressed in terminal ileum and large intestine [149,150,151]. Like SMCT1, SMCT2 mediates absorption of short-chain monocarboxylates, but with lower affinity compared to SMCT1 [151,152]. E23_ox_ reduced the expression of both *Slc5a8* (SMCT1) and *Slc5a12* (SMCT2), which might decrease the absorption of small-chain fatty acids by the energy substrate during EBOV infection [90].

Intestinal fluid secretion is mediated by various mechanisms in enterocytes of the small intestine. Na^+^/K^+^ ATPase transports three Na^+^ ions out of the cell through the basolateral membrane in exchange for two K^+^ ions that are pumped into the cell by using cellular energy in the form of ATP hydrolysis and contributes to the negative membrane potential [153]. Low concentration of intracellular sodium established by Na^+^/K^+^ ATPase drives the activity of the Na^+^/K^+^/2Cl^−^ cotransporter (NKCC1) that allows entry of sodium and potassium into the cell through the basolateral membrane, resulting in the activation of potassium channels such as KCNQ1 and KCNN4 that allow exit of K^+^ ions to prevent depolarization [154]. NKCC1 cotransporter also allows entry of chloride into cells that eventually exits the cells through the chloride channels in the apical membrane. The channels that mediate the efflux of chloride ions are cystic fibrosis transmembrane conductance regulator (CFTR) and Ca^2+^-activated chloride channels (CaCCs). Transepithelial chloride secretion results in paracellular Na^+^ secretion followed by passive passage of water due to an osmotic gradient established by transcellular passage of various electrolytes. RNA-seq analyses performed by Melnik and coworkers demonstrated that E23_ox_ downregulated the expression of *KCNQ1*, which encodes a potassium voltage-gated channel (KCNQ1) [90]. Tight junctions link adjacent epithelial cells and provide an intestinal barrier. This paracellular barrier controls selective transport of solutes and water through tight-junction pores. Claudins and occludins are proteins that form tight junctions [155]. E23_ox_ reduced the expression of *Cldn3*, *Cldn15*, and *Ocln*, which code for claudin-3, claudin 15, and occludin, respectively [90]. Downregulation of these tight-junction proteins would result in disruption of tight junctions and increases in passage of solutes and water between the cells, leading to diarrhea.

TMEM16A, also known as ANO1, is a Ca^2+^-dependent chloride channel (CaCC) [156,157]. Ca^2+^-dependent chloride secretion is defective in mice that do not express ANO1 protein [158,159,160]. ANO1 is highly expressed in mouse submandibular salivary gland acinar cells [161], and ANO1-mediated chloride secretion is necessary for normal mouse airway surface liquid homeostasis [162]. E23_ox_ induced the expression of *Ano1*, which codes for ANO1 protein [90]. To investigate the role of ANO1 in E23_ox_-induced fluid secretion, we used three ANO1 channel inhibitors: MONNA, CaCC_inh_ -A01, and T16A_inh_ -A01 [163,164]. These inhibitors have not been previously tested in vivo. MONNA has been shown to relax preconstricted murine mesenteric arteries and inhibit constriction in the presence and absence of chloride. Additionally, MONNA induced hyperpolarization of rat mesenteric arteries, presumably through activation of potassium conductance [165]. We tested MONNA at 10 μM and 20 μM together with 100 μM E23_ox_ in our murine ligated ileal loop model. MONNA at these concentrations was not very effective. Treatment of loops with 20 μM MONNA and 100 μM E23_ox_ resulted in a higher number of loops positive for E23_ox_-induced fluid secretion compared to those treated with MONNA at the concentration of 10 μM and 100 μM E23_ox_, suggesting that MONNA at higher concentrations induces fluid secretion, possibly through hyperpolarization of the cell membrane. E23_ox_ downregulated the expression of potassium voltage-gated channel KCNQ1. MONNA may activate E23_ox_-downregulated potassium channels or other potassium channels, resulting in hyperpolarization of the membranes and induction of fluid secretion. In contrast to the study performed by Boedtkjer et al., where they showed that CaCC_inh_-A01 inhibited constriction of noradrenaline- and U46619-stimulated mouse mesenteric arteries, in our study, treatment of loops with 10 μM CaCC_inh_-A01 and 100 μM E23_ox_ resulted in induction of fluid secretion in seven out of nine loops. The most effective inhibitor of E23_ox_-induced fluid secretion was T16A_inh_-A01. T16A_inh_-A01 showed inhibition in 8 out of 10 loops (80%). Our data suggest that ANO1 channel may play a role in E23_ox_-mediated fluid secretion.

Prostaglandin E2 (PGE2) mediates intestinal fluid secretion and has been shown to induce fluid secretion in jejunal and colonic organoids [166]. High levels of PGE2 were detected in plasma and stool of children with rotavirus gastroenteritis [167]. E23_ox_ upregulated the expression of *Ptgs2*, which codes for prostaglandin-endoperoxidase synthase-2 (COX-2), the enzyme that synthesizes prostaglandin precursors, suggesting an increase in the levels of PGE2 as the result of E23_ox_ treatment, suggesting the possible involvement of PGE2 in E23_ox_-induced diarrhea [90]. PGE2 is synthesized from arachidonic acid in cell membranes and interacts with one of four receptors (EP1-EP4) to exert its pleotropic effects. EP1 coupled to Gα_q_ activates PLCβ and initiates the IP_3_ cascade that results in increase in intracellular Ca^2+^ levels and activation of PKC and ANO1 [168,169]. ANO1 is a linker protein that tethers inositol triphosphate (IP_3_) receptors and intracellular Ca^2+^ to membrane channels, resulting in their activation [170]. EP1 has the least affinity to PGE2 [171] and is probably activated when COX-2 is upregulated and PGE2 levels are high. PGE2s bind EP2, EP4, and EP3γ coupled to Gα_s_, which leads to activation of adenylate cyclase (AC) and increase in cAMP levels [172]. PGE2 activates EP4 receptor and induces chloride secretion through the crosstalk between cAMP and Ca^2+^ signaling in mouse medullary collecting duct cells [173]. EP3 variants are principally coupled to Gα_i_, and their binding to PGE2 results in inhibition of AC and decrease in cAMP levels [174]. EP3 coupled to Gα_i_ can also activate PLCβ and IP_3_ signaling cascade, resulting in increase in Ca^2+^ levels and activation of PKC, similar to signaling initiated by EP1 coupled to Gα_q_ [175]. We tested the role of PGE2 in E23_ox_-induced fluid secretion in our in vivo model of diarrheal disease. Ligated ileal loops were injected with 100 μM indomethacin, a COX-2 inhibitor, together with 100 μM E23_ox_. Interestingly, ten out of eleven loops instilled with E23_ox_ and indomethacin and eight out of ten loops instilled with E23_ox_ and T16A_inh_-A01 were negative for fluid secretion, suggesting a possible involvement of ANO1 channel and PGE2s in E23_ox_-induced fluid secretion [90]. T16A_inh_-A01 and indomethacin treatments did not result in a complete inhibition of diarrheal response induced by E23_ox_, suggesting the complexity of the mechanism of diarrhea induced by EBOV delta peptide. EBOV delta peptide may permeabilize the ileal membrane, allowing chloride ion efflux that leads to fluid secretion [102,104].

## 4. Discussion

Viroporins are ion channels that change cellular ion homeostasis and are essential for viral replication and pathogenesis. They form tertiary structures consisting of several transmembrane domains and perturb target cellular membranes by pore formation, allowing movement of ions and resulting in disruption of membrane gradient and cellular homeostasis. Transmembrane domains of viroporins are usually amphipathic α-helices containing non-polar, polar, and positively charged amino acids, with non-polar residues located on one side of the helix and polar and positively charged residues on the opposite side. Oligomerization of transmembrane domains of viroporins is usually mediated by hydrophobic residues. Modeling studies of Ebola virus delta peptide predicted that delta peptide forms an amphipathic α-helix that could form either a tetrameric or pentameric tertiary structure. Garry and Gallaher [102] hypothesized that hydrophobic amino acids on one side of the amphipathic α-helix would interact with the lipid environment of the target membrane and positively charged amino acids lining the interior of the pore would allow ion flux. Simulation studies performed by Pokhrel and coworkers [103] showed that a disulfide bond is essential for pore formation and stabilization. Ebola virus delta peptide and STb are 40 and 48 amino acids in length, respectively, and have the same number of lysine (6K) and arginine (2R) residues. While EBOV delta peptide forms disulfide-linked hairpin structure with an amphipathic α-helix, STb monomer, on the other hand, consists of one amphipathic and one hydrophobic α-helix connected by two disulfide bonds. STb multimerizes into hexamers and heptamers through hydrophobic interactions between residues F37, I41, and M42 (Figure 3B,C), where the hydrophobic α-helices are probably on the exterior of the toxin-facing lipids of the cellular membrane and amphipathic α-helices coating the interior of the STb pore. Future structural studies of EBOV delta peptide are needed to determine a tertiary structure of delta peptide and to identify the residues that are essential in the formation of the oligomer.

Viroporins form pores that are often characterized by weak ion selectivity. This characteristic of viroporins allows them to achieve various biological functions. EBOV delta peptide has been shown to form small pores and change ion permeability across cell membranes [104]. All-atom molecular simulations showed that delta peptide is selective to chloride ion and does not allow passage of sodium [103]. Future studies in cell systems are needed to determine ion selectivity of Ebola virus delta peptide, which might open new venues to determine other possible roles of delta peptide in the replication cycle of EBOV. STb, instead, forms nonspecific pores [42], allowing different ions to pass through the membrane. The nonselective nature of pores made by STb indicates that the mechanism of action of this toxin may not be complex.

Intestinal fluid secretion as a hallmark of diarrheal disease has been studied using suckling mouse assay and ligated ileal loop assay. The majority of STb diarrheal studies have been performed by employing ligated ileal loop assay in different animal models including neonatal pigs, weaned pigs, rats, rabbits, and lambs. The maximal diarrheal response was observed 2–6 h after STb treatment. The peak of delta peptide’s enterotoxigenic activity, on the other hand, is 9–12 h after injection into ileal loops of mice. STb is sensitive to degradation by trypsin found in the intestine. Consistent results due to STb treatment were observed only in weaned pigs, and it was initially believed that STb is a pig-specific toxin. Saline rinsing of jejunum before STb inoculation and protease inhibitor treatment blocked endogenous protease activity and produced consistent STb-induced diarrheal response in other animals. EBOV delta peptide produced diarrhea in approximately 60% of mice tested. Trypsin was previously shown to target lysine and arginine residues [49]. EBOV delta peptide and STb have the same number of lysine and arginine residues (6K and 2R). It is possible that the inconsistent EBOV delta peptide results in the murine model of diarrheal disease were due to endogenous protease degradation of EBOV delta peptide in mouse ileum. The majority of residues important in enterotoxigenic activity of STb are positively charged with a smaller number of hydrophobic residues and one negatively charged residue (Table 1 and Table 2). Our in vitro studies using Vero cells demonstrated that cysteines and positively charged residues are essential for cytotoxic activity of EBOV delta peptide. We have not tested delta peptides with these modifications in our murine ligated ileal loop model. It is plausible that they would induce a less severe diarrheal response. While the loss of a single disulfide bond that did not affect the secondary structure of STb was sufficient to abolish STb-induced diarrhea in rat intestinal loop assay [30], disulfide bond cleavage due to sulfhydryl alkylation of EBOV delta peptide still permitted moderate induction of fluid secretion, suggesting the importance of delta peptide’s secondary structure in the induction of diarrhea (Table 2) [90].

Pathogen-induced fluid secretion is accompanied by histological changes and damage to the intestinal mucosa. The extent of the cellular and structural alterations of the epithelium that lines the small intestine can range from very minimal to more dramatic and correlate with the absorptive function of the intestine. Treatment with STb induced minimal changes to the intestinal mucosa that include partial villous atrophy, shortening of villi, crypt hyperplasia, and sloughing of the cells. Cellular damage was not observed. These morphological changes induced by STb imply that the number of absorptive cells responsible for absorption of ions and nutrients is reduced. Absorption of alanine in some STb-treated porcine jejunal loops was reduced, but it was independent of the histological alterations induced by the toxin [176]. EBOV delta peptide induced more severe damage to the intestinal mucosa compared to the changes caused by STb. These alterations include maximal distension of the loops that was accompanied by extreme thinning of the basal lamina, loss of villi, sloughing of cells, cellular damage, and increase in the crypt length. RNA-seq analyses demonstrated that EBOV delta peptide downregulates the expression of channels responsible for absorption of ions, sugars, and fatty acids in mouse intestine. In-depth functional studies of channels will show the extent to which their function is altered due to EBOV delta peptide and could provide insight into management of gastrointestinal disease associated with EBOV infection.

The pathogenic mechanisms of viral and bacterial enterotoxins are complex and, in the cases of STb and EBOV delta peptide, not yet fully elucidated. Common mechanisms of induction of diarrhea used by STb and Ebola virus delta peptide are opening of tight junctions and increase in prostaglandin synthesis (Figure 2 and Table 2). STb treatment resulted in redistribution and fragmentation of tight-junction proteins, as well as their displacement to the cytoplasm [66,67]. Similarly, E23_ox_ downregulated the expression of tight-junction proteins [90]. EBOV delta peptide upregulated the expression of COX-2, and treatment with COX-2 inhibitor indomethacin inhibited diarrhea induced by delta peptide [90]. STb has been shown to increase the release of arachidonic acid, resulting in prostaglandin synthesis. Serotonin release as the result of STb treatment was also observed and has been shown to be partially responsible for prostaglandin synthesis [52,62,63]. Increase in serotonin levels due to STb treatment indicates the involvement of the enteric nervous system, which has been also shown in rotavirus-induced diarrheal disease [177]. The role of serotonin in EBOV delta peptide-induced diarrhea has not been investigated yet, but research in this area could be of interest in the future. Both STb and EBOV delta peptide induce diarrhea through Ca^2+^ signaling. Delta peptide has been shown to engage ANO1, a Ca^2+^-activated chloride channel (Table 2) [90]. STb, on the other hand, induces the increase in Ca^2+^ in the cytosol that leads to the activation of CAMKII [61]. More studies are required to show whether STb engages CFTR or CaCC for induction of fluid secretion (Table 2). It has been shown that STb induces diarrhea in a adenylate-cyclase (AC)-independent pathway [52,62]. Similarly, delta peptide was shown to downregulate AC, implying that delta peptide does not use cAMP as a second messenger in induction of fluid secretion (Figure 2) [90]. Recent identification of EBOV delta peptide as an enterotoxin opens multiple venues for future studies to potentially elucidate the mechanism of induction of gastrointestinal disease during EVD.

## Figures and Tables

**Figure 1 pathogens-11-00170-f001:**
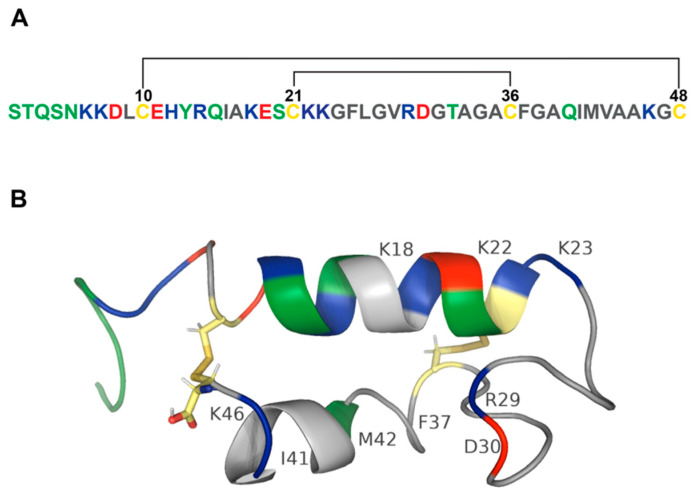
Structure of STb toxin. (**A**) Amino acid sequence of STb. Polar amino acids (serine, threonine, tyrosine, asparagine, and glutamine) are colored green. Positively charged amino acids (lysine, arginine, and histidine) are blue. Negatively charged amino acids (aspartic acid and glutamic acid) are red. Nonpolar amino acids (glycine, alanine, valine, proline, leucine, isoleucine, methionine, tryptophan, and phenylalanine) are grey. Cysteines are colored yellow. STb contains two disulfide bonds schematically depicted by black lines. (**B**) Secondary structure of STb. A model of the 48-amino-acid peptide sequence of STb was rendered in PyMol [29,37,38]. PDB structure of STb: 1EHS. STb toxin contains two α-helices connected by two disulfide bonds. Amphipathic α-helix consists of basic (blue), polar (green), acidic (red), and hydrophobic (grey) amino acids and is separated from the hydrophobic α-helix by a linker sequence consisting of mostly hydrophobic (grey) residues and a pair of charged amino acids. Disulfide bonds between cysteines are colored by element (hydrogen—white; nitrogen—blue; oxygen—red; sulfur—gold). Labeled residues play an important role in STb pathogenesis.

**Figure 2 pathogens-11-00170-f002:**
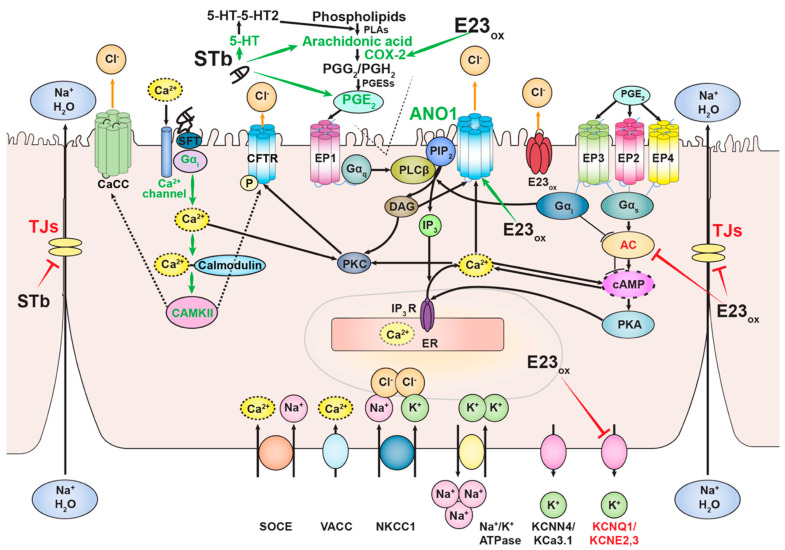
Mechanisms of action of Ebola virus delta peptide and STb. Both toxins induce the synthesis of prostaglandin (PGE2) in the plasma membrane and inhibit tight-junction (TJ) proteins that link adjacent enterocytes in the intestinal mucosa. Ebola virus delta peptide upregulates the expression of COX-2, resulting in increase in PGE2 synthesis. Delta peptide causes diarrhea through PGE2 signaling and Ca^2+^-dependent chloride channel, ANO1. Binding of STb to its receptor, sulfatide (SFT), is followed by activation of GTP-binding protein and influx of extracellular Ca^2+^ through the Ca^2+^ channel located in the plasma membrane. Increase in cytosolic Ca^2+^ levels results in the formation of Ca^2+^-calmodulin complex and activation of calmodulin-dependent protein kinase II (CAMKII). CAMKII potentially activates either cystic fibrosis transmembrane regulator (CFTR) by phosphorylating the channel or unidentified Ca^2+^-activated chloride channel (CaCC) that leads to STb-induced chloride flux into the lumen, resulting in fluid secretion (dashed arrows). STb induces PGE2 synthesis through serotonin (5-HT) activation and arachidonic acid metabolism. Activation and inhibition of proteins by both toxins are depicted by green and red arrows and text, respectively.

**Figure 3 pathogens-11-00170-f003:**
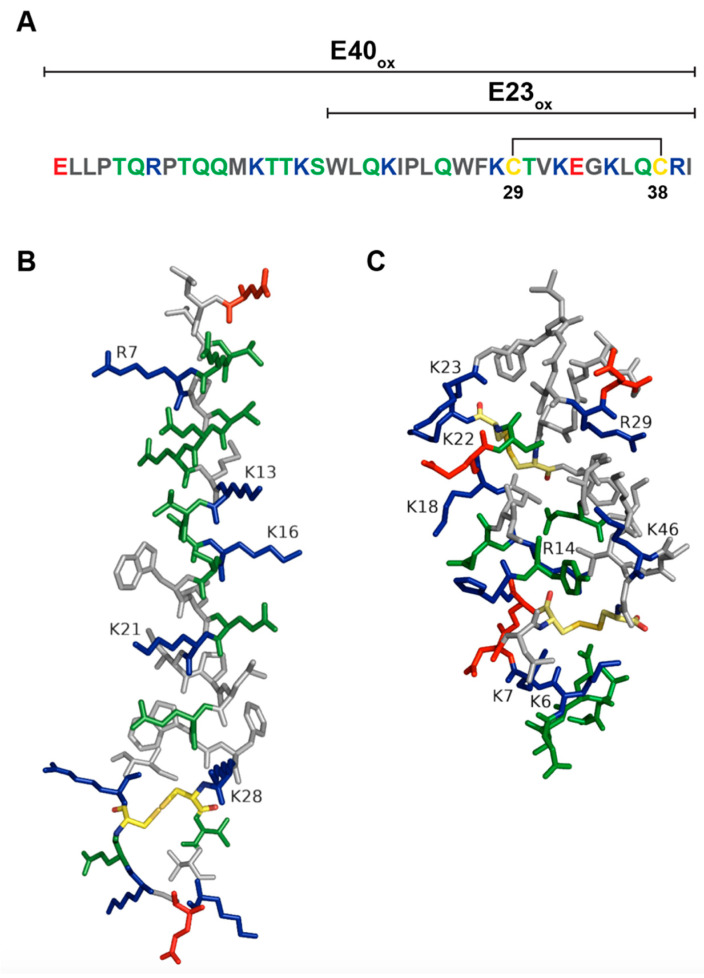
Ebola virus delta peptide. (**A**) Amino acid sequence of Ebola virus delta peptide. Polar amino acids (serine, threonine, tyrosine, asparagine, and glutamine) are colored green. Positively charged amino acids (lysine, arginine, and histidine) are blue. Negatively charged amino acids (aspartic acid and glutamic acid) are red. Nonpolar amino acids (glycine, alanine, valine, proline, leucine, isoleucine, methionine, tryptophan, and phenylalanine) are grey. Cysteines are colored yellow. Disulfide bond is indicated by black lines. (**B**) A model of Ebola virus delta peptide. Amino acid sequence of Ebola virus delta peptide (EBOV/SLE14-EM95) was rendered in PyMol [38] and Adobe Illustrator. (**C**) STb structure was rendered in PyMol [38]. PDB structure of STb: 1EHS.

**Table 1 pathogens-11-00170-t001:** Role of STb residues in pathogenesis of the toxin.

Amino Acid	Enterotoxicity	Receptor Binding	Oligomerization	Pore Formation	TJ Opening
C10	•		•	•	
K18	•				
C21	•		•	•	
K22	•	•		•	
K23	•	•		•	
R29	•	•			
D30	•	•			•
C36	•		•	•	
F37	•	•	•		
I41	•	•	•		
M42	•	•	•	•	
K46	•				
C48	•		•	•	

• denotes activity.

**Table 2 pathogens-11-00170-t002:** Requirements for the induction of diarrhea by EBOV delta peptide and ETEC STb.

	Cationic Residues	α-Helices	Disulfide Bonds	Chloride Channels	PGE2 Synthesis	Calcium Signaling	TJ Opening
ETEC STb	•		•	•	•	•	•
EBOV delta peptide		•		•	•	•	•

• denotes activity.

## Data Availability

Not applicable.

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
