# Peer review of "Enterotoxigenic Escherichia coli Heat-Stable Toxin and Ebola Virus Delta Peptide: Similarities and Differences"

_pathogens, 2022, doi:10.3390/pathogens11020170_

Round 1
Reviewer 1 Report
In this review the authors compare structural and functional properties of two small (40-48 amino acid) proteins: the STb pore forming toxin from Enterotoxigentic E. Coli (ETEC) and the delta peptide from ebolavirus. Both proteins have been associated with diarrheal disease.
Specific Comments
- Line 97: provide a brief description of sulfatide (eg, sulfated glycosylceramide found in cell plasma membranes (If accurate))
- Table 1: For completeness the authors may wish to add information on the 4 Cys residues
- Line 125-126: “leading to mitochondrial hyperpolarization” (?)
- Lines 127-128: “Fig. 1B and Table 1 in [38]”(?)
- L 187: specify if the key glycoside is on the receptor, or on the toxin (presumably on the sulfatide receptor)
- L192: “a Ca+2-calmodulin complex” (?)
- Line 210: Do you mean (current) Fig. 3
- If so to above comment, perhaps swap order of current Figs 2 and 3.
- Current Fig. 2: It might be useful to also present a version of the delta peptide model as a Ribbon diagram (to match STb in Fig. 1).
- Several sentences need to be checked/corrected for sentence structure (e.g., Lines 248-249, 261-262, 265, 304, 336, etc).
- Line 287: “supported” (vs “confirmed”; as stated, NMR or x-ray data are needed to confirm the structural model for delta peptide)
- Line 290: Do you mean “tertiary structure” (vs. “secondary structure”)
- Line 290-291: Provide a citation or more detail for the statement that the delta peptide has a high propensity to bind to membranes. Is this propensity based on experimental or computer modeling data.
- A citation should be provided for Melnik et al. as it forms the basis for much of the text regarding the ebolavirus delta peptide, which accounts for about half of the review. Is this a peer-reviewed or recently submitted manuscript or a manuscript in preparation?
- Line 298: It would be more appropriate to say “could form” (vs. “forms”)
- Line 318: “These data…”
- Line 321: “transmembrane domains” or should it read “amphipathic helices”?
- Lines 339-341 (and Ref. 112): state the species demonstrated in
- Line 360: “high predicted structural similarity”
- RE: abbreviations: (a) What does DRA abbreviate? (It seems to be used to denote several channels/transporters.) (b) Line 528: Define AC (adenylate cyclase) here, at its first use.
Author Response
Comments and Suggestions for Authors
In this review the authors compare structural and functional properties of two small (40-48 amino acid) proteins: the STb pore forming toxin from Enterotoxigentic E. Coli (ETEC) and the delta peptide from ebolavirus. Both proteins have been associated with diarrheal disease.
Specific Comments
- Line 97: provide a brief description of sulfatide (eg, sulfated glycosylceramide found in cell plasma membranes (If accurate)).
Response: We added a description of the STb receptor with 2 references.
Labrie and coworkers studied protein-protein interaction of STb toxin and reported that STb forms hexamers and heptamers in a temperature- and receptor-independent manner [34]. The functional receptor for STb is the sulfatide [Gal(3-SO4)b1Cer], a glycosphingolipid that is present on the cell surface of the jejunal epithelium [35,36].
- Table 1: For completeness the authors may wish to add information on the 4 Cys residues
Response: Thank you for pointing out this important detail. Table 1 has been updated with the addition of four cysteines and their role in STb’s pathogenesis.
- Line 125-126: “leading to mitochondrial hyperpolarization” (?)
Response: Thank you for the suggestion to clarify the findings of this study. We have
added the explanation for the importance of mitochondrial hyperpolarization and added
two more references of previous studies that have shown the link between mitochondrial
alterations and apoptosis.
STb permeabilized membranes of NIH-3T3 cells and was internalized by the cells.
Initially, STb was found scattered throughout the cytoplasm and later in was detected in
mitochondria that lead to mitochondria hyperpolarization [42], which might be an
indication of early stages of apoptosis as shown by previous studies [43,44]. Treatment
with STb resulted in histological changes such as cytoplasm granularization and
morphological alterations associated with apoptosis including enlargement of the
nucleus and membrane budding [42].
- Lines 127-128: “Fig. 1B and Table 1 in [38]”(?)
Response: We agree with the reviewer’s suggestion, and we moved “(Figure 1B and
Table 1)” in the end of the sentence that describes the importance of STb residues in
STb’s pathogenesis.
The disulfide bridges and residues K22, K23, and M42 that are essential for STb’s enterotoxicity
and oligomerization of STb are also indispensable for the ability of STb to permeabilize cellular
membranes (Figure 1B and Table 1) [41].
- L 187: specify if the key glycoside is on the receptor, or on the toxin (presumably on the sulfatide receptor)
Response: We added this important clarification in text.
STb binds to its receptor, sulfatide, through terminal oligosaccharide sequence [Gal(3-
SO4)b1Cer] on sulfatide with the optimum binding pH of 5.8 [35,36,56].
- L192: “a Ca+2-calmodulin complex” (?)
Response: Calmodulin is an intracellular Ca2+ receptor that mediates Ca2+ regulation of
glycogen metabolism, secretion, motility, and Ca2+ transport. Calmodulin is highly
conserved and binds Ca2+ with high affinity and specificity. The binding of Ca2+ to
calmodulin is illustrated in Figure 2. We also added this detail in text.
Intracellular increase of Ca2+ levels in response to STb leads to binding of Ca2+ to its
receptor calmodulin, which results in the formation of Ca2+- calmodulin complex that
promotes activation of calmodulin-dependent protein kinase II (CAMKII) [61].
- Line 210: Do you mean (current) Fig. 3
Response: Thank you for noting this mistake. Yes, we were referring to current Figure 3.
- If so to above comment, perhaps swap order of current Figs 2 and 3.
Response: We swapped the figures 2 and 3.
- Current Fig. 2: It might be useful to also present a version of the delta peptide model as a Ribbon diagram (to match STb in Fig. 1).
Response: Unfortunately, it is not possible due to the fact the crystal structure of Ebola
virus delta peptide has not been solved yet.
- Several sentences need to be checked/corrected for sentence structure (e.g., Lines 248-249, 261-262, 265, 304, 336, etc).
Response: Thank you for pointing out these mistakes. We corrected these sentences.
- Line 287: “supported” (vs “confirmed”; as stated, NMR or x-ray data are needed to confirm the structural model for delta peptide)
Response: We changed “confirmed” to “supported”.
- Line 290: Do you mean “tertiary structure” (vs. “secondary structure”)
Response: In this sentence we were referring to the secondary structure of EBOV delta peptide. While analyzing the sequence of the peptide, we were predicting the biologically-active secondary structure of delta peptide.
- Line 290-291: Provide a citation or more detail for the statement that the delta peptide has a high propensity to bind to membranes. Is this propensity based on experimental or computer modeling data.
Response: We clarified this statement by adding more detail on the method used and added appropriate references.
EBOV delta peptide was also tested for its propensity to interact with lipid membranes by using
the Wimley-White interfacial hydrophobicity scale (WWIHS), an experimentally determined
hydrophobicity scale that predicts regions of proteins that may interact with membranes [101].
EBOV delta peptide had a positive score of 3.66, which indicates that it has a high propensity to
interface with lipid membranes [102].
- A citation should be provided for Melnik et al. as it forms the basis for much of the text regarding the ebolavirus delta peptide, which accounts for about half of the review. Is this a peer-reviewed or recently submitted manuscript or a manuscript in preparation?
Response: Melnik et al. has been accepted upon submission of this manuscript and was
published in Cell Reports on January 4, 2022. We provided the references in the current
version of the text.
- Line 298: It would be more appropriate to say “could form” (vs. “forms”)
Response: We changed “forms” to “could form”.
- Line 318: “These data…”
Response: We changed “this data” to “these data”.
- Line 321: “transmembrane domains” or should it read “amphipathic helices”?
Response: We changed “transmembrane domains” to “amphipathic helices”.
- Lines 339-341 (and Ref. 112): state the species demonstrated in
Response: We added the species (mice).
Ball and coworkers demonstrated that NSP4 and a synthetic peptide corresponding to residues 114 to 135 of the protein induce diarrhea in mice in an age-dependent and dose-dependent manner [118].
- Line 360: “high predicted structural similarity”
Response: We changed “high structural similarity” to “high predicted structural similarity”.
- RE: abbreviations: (a) What does DRA abbreviate? (It seems to be used to denote several channels/transporters.) (b) Line 528: Define AC (adenylate cyclase) here, at its first use.
Response: We defined DRA, a down-regulated in adenoma. Adenylate cyclase (AC) was defined in line 524 and was abbreviated in line 528.

Reviewer 2 Report
The authors present a comprehensive review of the structural, functional, and pathological similarities and differences between E. coli STs and EBOV delta-peptide in the context of gastrointestinal pathology in the respective disease states caused by each agent. The review is well written, appropriately referenced, and highlights an important aspect of the pathogenesis of both agents. I found no causes for concern.
Author Response
Comments and Suggestions for Authors
The authors present a comprehensive review of the structural, functional, and pathological similarities and differences between E. coli STs and EBOV delta-peptide in the context of gastrointestinal pathology in the respective disease states caused by each agent. The review is well written, appropriately referenced, and highlights an important aspect of the pathogenesis of both agents. I found no causes for concern.
Response: We thank the reviewer for reading the manuscript and providing us with the feedback.